# Lipase Addition Promoted the Growth of *Proteus* and the Formation of Volatile Compounds in *Suanzhayu*, a Traditional Fermented Fish Product

**DOI:** 10.3390/foods10112529

**Published:** 2021-10-21

**Authors:** Cuicui Jiang, Mengyang Liu, Xu Yan, Ruiqi Bao, Aoxue Liu, Wenqing Wang, Zuoli Zhang, Huipeng Liang, Chaofan Ji, Sufang Zhang, Xinping Lin

**Affiliations:** National Engineering Research Center of Seafood, Collaborative Innovation Center of Provincial and Ministerial Co-Construction for Seafood Deep Processing, Liaoning Province Collaborative Innovation Center for Marine Food Deep Processing, School of Food Science and Technology, Dalian Polytechnic University, Dalian 116034, China; jcc07061013@foxmail.com (C.J.); morisaliu@hotmail.com (M.L.); yanxu8088@gmail.com (X.Y.); rickibao@sohu.com (R.B.); liuaoxue.lax@gmail.com (A.L.); wangwenqing1104@gmail.com (W.W.); zzl2632098660@gmail.com (Z.Z.); lianghp@dlpu.edu.cn (H.L.); jichaofan@outlook.com (C.J.); zhangsf@dlpu.edu.cn (S.Z.)

**Keywords:** fermented fish, *Proteus*, lipase, volatile compounds, aldehydes, esters

## Abstract

This work investigated the effect of lipase addition on a Chinese traditional fermented fish product, *Suanzhayu*. The accumulation of lactic acid and the decrease of pH during the fermentation were mainly caused by the metabolism of *Lactobacillus*. The addition of lipase had little effect on pH and the bacterial community structure but promoted the growth of *Proteus*. The addition of lipase promotes the formation of volatile compounds, especially aldehydes and esters. The formation of volatile compounds is mainly divided into three stages, and lipase had accelerated the fermentation process. *Lactobacillus*, *Enterococcus* and *Proteus* played an important role not only in inhibition of the growth of *Escherichia-Shigella*, but also in the formation of flavor. This study provides a rapid fermentation method for the *Suanzhayu* process.

## 1. Introduction

Fermentation is a traditional preservation technique that provides a unique aroma and nutritional values with the action of microorganisms or endogenous enzymes. *Suanzhayu* is a type of traditional solid fermented fish in China, which is produced by mixing rice powders with seasonings and fresh fish meat in a sealed fermentation condition. Due to the metabolism of microorganisms during the fermentation process, the product develops a unique sour aroma; thus, it is popular among the consumers [1]. In recent years, the effects of a starter culture on *Suanzhayu’s* quality, safety, bacterial community structure, and base flavor formation have been extensively studied [1]. However, the process still involves a long fermentation period. Moreover, it is difficult to control the changes of microbial community and flavor during fermentation, which often leads to unstable product quality. Therefore, the addition of enzymes, such as protease and lipase to shorten the fermentation maturation process has attracted the attention of many researchers [2,3].

Lipase increases the content of free fatty acids through hydrolysis reactions. Free fatty acids can be converted to volatiles such as aldehydes and ketones by microbial enzymatic or non-enzymatic reactions in the system [4], which increase the flavor of the product. Many papers have reported that lipases are associated with the formation of characteristic food flavors. For example, the lipolysis of lipase is essential for the characteristic flavor of food, such as that in the mature white cheese flavor [5]. Researchers found out that lipase addition improved the desirable dairy volatile compounds, such as butyric acid, hexanoic acid, 2-nonanone, and hexanal resulting from lipolysis in milk [6]. In addition, it has been reported that lipase addition can promote the formation of antibacterial flavor substances, such as octanal and nonanal, thereby inhibiting the growth of citrus *Geotrichum citri-aurantii* mycelium [7]. However, thus far, there are no studies on the relationship between lipase on bacterial community succession, physicochemical indicators and changes in aroma properties of *Suanzhayu*.

Based on the previous studies, we hypothesized that lipase addition might improve the aromas, such as aldehydes and esters during *Suanzhayu* fermentation. Therefore, in this study, the effect of lipase addition on *Suanzhayu* fermentation was investigated. The correlation between bacterial community succession and the changes of *Suanzhayu* product properties, such as pH, lactic acid content, and volatile components, were analyzed and discussed.

## 2. Materials and Methods

### 2.1. Preparation of Suanzhayu Samples

*Suanzhayu* samples were purchased from a local Qianhe Market (Dalian, Liaoning, China) using fresh carp (*Cyprinus carpio* L.) with an average weight of 2 ± 0.2 kg. The fish were gutted, cleaned, and cut into cubes of 2.5 × 2.5 × 2.5 cm.

The lipase addition is based on the following reasons. First, the lipase activity (PANGBO ENZYME, Nanning, China) is 100,000 U/g, and the recommended dose is 0.1 to 0.3%, i.e., 100 U/g to 300 U/g. Second, 100 to 300 U/g is a common addition in fermented products [8]. For example, Rani and Jagtap reported that 200 U/g of lipase was added to cheese, and the cheese maturation time was shortened from 90 days to 60 days while maintaining its quality [8]. Finally, different lipase addition at 100, 200 and 300 U/g were prepared, and the pre-experiment showed that the effect of adding 100 U/g fish meat of lipase was of good acceptance. Therefore, we decided to add 100 U/g lipase in *Suanzhayu*. Two groups were prepared: (i) group U100, in which 100 U/g fish meat of lipase was added; and (ii) group U0, in which the same volume of sterile water was added. Each fermented jar was added with 200 ± 5 g of fish meat, mixed with 3% salt, 30% rice flour and the enzyme solution. Then, the mixtures were placed in completely sealed jars and kept at 25 °C for 14 d. Samples was performed in triplicate from independent jars incubated for 0, 3, 5, 7, and 14 d. Samples for volatile compounds and microbiological analysis were kept at −80 °C and samples for other analysis were stored at −20 °C. Subsequent analysis was carried out as soon as possible within two months.

### 2.2. Determination of pH and Lactic Acid Contents

Sterile water (20 mL) was added into 2 g of sample, and the mixture was homogenized (4 × 15 s at 8000 rpm), (T25 digital ULTRA TURRAX^®^, IKA, Staufen, Germany). The supernatant was then subjected to pH measurement using a pH meter (FE28, Mettler Toledo, Greifensee, Switzerland). The lactic acid content was determined according to the previous method [9]. Briefly, 0.1 g sample was thoroughly minced with 10 mL sterile water. Lactic acid was extracted by ultrasonic for 10 min. The supernatant was filtered through a 0.45 μm membrane filter, following by a Cleaner SC18 SPE column. The lactic acid content was determined with high-performance ion chromatography (Dionex ICS-5000 + DC, Thermo Scientific, Waltham, MA, USA) with the same chromatographic condition described by Lv et al. [9]. All the experiments were carried out in triplet and expressed as the mean ± standard error.

### 2.3. Analysis of Bacterial Community

*Suanzhayu* was aseptically sampled from the fermented jar and stored at −80 °C, and then samples were sent to the Biomarker Bioinformatics Technology Co., Ltd. (Beijing, China) for 16S rRNA gene amplicon. The total genomic DNA was extracted from a 0.25 g sample with EZNA^®^ DNA kit (Omega, Norcross, GA, USA). The hypervariable regions V3 and V4 of 16S rRNA were amplified using special primers 338F (5’-ACTCCTACGGGAGGCAGCA-3’) and 806R (5’-GGACTACHVGGGTWTCTAAT-3’). After amplification and purification of the fragments, the gene library was established using DNA Library Preparation Kit (Ade Technology Co., LTD, Beijing, China) and was subjected to sequencing using an Illumina HiSeq 2500 platform.

Paired-end reads were assembled using FLASH version 1.2.7 to obtain raw tags, which were then filtered based on quality using Trimmomatic version 0.33. The chimera tags were then removed using UCHIME version 4.2, and the effective tags were obtained. The effective tags with a similarity of more than 97% were clustered, and operational taxonomic units (OTUs) were determined using UPARSE software. For each OTU, a representative sequence with a high frequency of occurrence is classified by searching the Silva database using the Ribosomal Database Project (RDP) classifier version 2.2. The raw data have been submitted to the National Centre for Biotechnology Information (NCBI) website (Accession number: SRP230170).

### 2.4. Volatile Compounds Analysis by HS-SPME-GC-MS

Volatile compounds in the sample were determined by HS-SPME-GC-MS (Agilent Technologies, Santa Clara, CA, USA) equipped with HP-5MS capillary column (30 m × 250 μm × 0.25 μm) (Agilent, USA). *Suanzhayu* was steamed for 20 min and then minced thoroughly. Samples (2 g) were placed in a headspace extraction vial (20 mL, 18 mm) and cyclohexanone (Aladdin, American) (50 mg/L) were added to each sample (0–3 d 20 μL, 5–7 d 40 μL, 14 d 60 μL) as the internal standard. The volatile compounds were extracted with a solid-phase micro-extraction needle with divinylbenzene/Carboxen/polydimethylsiloxane (PDMS) fiber for 40 min. The GC-MS parameters were set according to Bao’s study. The volatile compounds were thermally desorbed at 250 °C for 5 min. The initial oven temperature kept at 30 °C for 5 min, and then reached to 50 °C at a rate of 3 °C/min and kept for 3 min. After that, the oven temperature was raised to 150 °C at a rate of 5 °C/min, following by raising to 250 °C at a rate of 20 °C/min (held for 5 min). The ionization source with the energy of 70 eV at 230 °C was used, and the mass scan range was 40–400 mass unit with an emission current of 150 μА. The retention index (RI) of n-alkanes (C7–C30, Sigma-Aldrich, St. Louis, MO, USA) under the same GC conditions as Bao’s study was calculated [10], and the volatile compounds were identified by comparing the calculated RI and the mass spectra of fragments in the NIT14 library. The peak area of each compound was used to semi-quantify the concentration of volatile compounds in the sample. Then, the odor activity value (OAV) was calculated based on OAV = C/OT, where C was the concentration of volatile compounds, and OT was the odor threshold, which was obtained from the literature [11]. All samples were conducted in triplicate.

### 2.5. Statistical Analysis

Statistical analysis was analyzed by one-way ANOVA and Spearman correlation using SPSS (version 22.0, IBM, Armonk, NY, USA) software. When *p* < 0.05, the data was considered to have a significant difference. The histograms were conducted using Origin 8.5 (Origin Lab Corp., Northampton, MA, USA). The heat maps were carried out by using R Studio (version 3.4.4). The heat maps of Pearson correlation coefficient were performed by TBtools (version 0.6652).

## 3. Results and Discussion

### 3.1. Changes of pH and Lactic Acid Contents during Fermentation

The changes in pH and lactic acid content were shown in Figure 1. The pH dropped rapidly in the first 7 d and slowed to 4.4 at the end of the fermentation. According to Zeng’s study, *Suanzhayu* with a pH lower than 4.6 is generally considered as mature [12,13]; thus, both of the groups almost went on the same fermentation process. Additionally, the lactic acid content gradually increased from 2.7 to 7.6 and 6.8 mg/kg in U0 and U100 groups, respectively. The culture pH decreases with the increase in the concentration of lactic acid. This accumulation of lactic acid during the fermentation process can inhibit the growth of spoilage and pathogenic bacteria, thereby extending the shelf life of the product [14]. However, the lactic acid content of the U100 group at the end of fermentation was lower than that of the U0 group. Sun [15] and Knez [16] reported that lipase can catalyze the esterification of lactic acid and other fatty acids with alcohol. Therefore, the lower lactic acid content in U100 group might be due to the consumption of lactic acid by lipase-catalyzed esterification. This may also provide the possibility of flavor enhancement in the enzyme addition group. In all, the results indicate that the addition of lipase had no effect on the pH of *Suanzhayu*.

### 3.2. Microbial Succession during Fermentation

The sequencing coverage rate was above 99.8% (Appendix A), suggesting that the sequencing depth was sufficient to reflect the composition of the bacterial ecosystem in *Suanzhayu* samples. The sequences with a similarity of above 97% were clustered, resulting in 383 OTUs.

As for the phylum level (Figure 2a), at day 0, Firmicutes, Proteobacteria, and Bacteroidetes were the dominant phyla with a relative abundance of 28.8%, 26.3%, and 19.4%, respectively. With the extension of fermentation time, Firmicutes and Proteobacteria rapidly grew, and their relative abundance increased to more than 98.0%, which inhibited the growth of other phyla. Firmicutes in the U0 group became the only dominant phylum in the final product (71.42%). In contrast, the dominant bacteria in the final product of the U100 group were Firmicutes (58.91%) and Proteobacteria (40.39%). The results suggest that the addition of lipase could influence the distribution of bacterial communities in *Suanzhayu* at phylum level.

At the genus level (Figure 2b), *Escherichia-Shigella* was the dominant bacterium on day 0 with a relative abundance of 17.1%. *Escherichia-coli* is a spoilage bacterium found in most fermented foods. It can lead to the formation of harmful biogenic amines in fermented meat and fish products [17]. The relative abundance of *Lactobacillus* increased rapidly from 3 days of fermentation, from 2.08% to 25.35% (U0) and 31.64% (U100), respectively. As the fermentation proceeded, the abundance of *Lactobacillus* remained basically stable, and became the dominant genus. Compared with the U0, the abundance of *Proteus* in U100 increased significantly (*p* < 0.01), especially after 7 days of fermentation, reaching 17.69%. It was inferred that the addition of lipase could promote the growth of *Proteus*. *Proteus*’ growth was reported to facilitate flavor formation. In a cheese model, *Proteus vulgaris* was found to be closely related with 3-methyl-1-butanol and more volatile aroma substances, such as aldehydes and esters, were detected [18].

The cluster analysis of the two groups was shown in Figure 2c. According to the results, *Suanzhayu* samples could be divided into three categories: (1) samples 0 d and U0-7 d; (2) samples U03-5 d and U100-3-7 d; and (3) samples U0-14 d and U100-14 d. The above results show that the fermentation process of the *Suanzhayu* sample mainly goes through three stages, early fermentation, middle fermentation and late fermentation. In addition, from the clustering results of U0 group and U100 group, it can be seen that the three stages of the two groups are relatively synchronized.

### 3.3. Volatile Components Generated during Fermentation

A total of 21 volatile compounds were detected in *Suanzhayu* samples (Appendix A). Only 11 volatile compounds were found in the day 0 samples, with a total of 9327.4 μg/kg of volatile flavor compounds, indicating that the rest of the compounds were produced during fermentation. The content of volatile flavor compounds continued to accumulate as fermentation proceeded. At the end of fermentation, the content of volatile compounds in the U100 group (18,196.9 μg/kg) was significantly higher than that in the U0 group (15,428.5 μg/kg). Moreover, the contents of aldehydes and esters were significantly increased in the U100 group, which were 1.4 and 11.1 times higher than those in the U0 group, respectively. The above results indicate that the addition of lipase for fermentation could promote the formation of volatile substances (especially aldehydes and esters) in *Suanzhayu*.

Alcohols were one of the important components of volatile compounds contributing to the aroma of *Suanzhayu.* Only two alcohols, 1-hexanol and 1-octene-3-alcohols, were detected in unfermented samples. This suggests that most of the volatile alcohols detected in *Suanzhayu* were produced by fermentation. During the fermentation process, both of the content (Appendix A) and percentage (Appendix A) of alcohols in the U0 group continued to increase, while the U100 group showed a trend of first increase and then decrease. This may be the result of the conversion of alcohols and acids (e.g., fatty acids, etc.) into esterification by microorganisms in the system [19]. This also coincided with the increase in the content of ethyl ester hexanoic acid and ethyl ester octanoic acid in the U100 group (Appendix A). Some specific alcohols, for example, 3-methyl-1-butanol, was found to be 3.5 times higher in the U100 group than in the U0 group. It has been reported that 3-methyl-1-butanol is produced by the auto-oxidation of unsaturated fatty acids, which plays an important role in the faint aroma (e.g., mushroom or metallic taste) of fresh seafood [20]. Therefore, the addition of lipase may have increased the fatty acids, precursors of 3-methyl-1-butanol, and thus played a positive role in the formation of product aroma. At the end of fermentation, there was no significant difference (*p* < 0.05) in the alcohol content between U0 and U100 samples, which were 6857.8 μg/kg and 6870.9 μg/kg, respectively, indicating that the addition of lipase had little effect on the total alcohol content of *Suanzhayu*. Therefore, the addition of lipase had little effect on the total amount of alcoholic substances in *Suanzhayu* but promoted its characteristic flavor substances, such as 3-methyl-1-butanol.

Aldehydes were also one of the main volatile components of *Suanzhayu*. At the end of fermentation, the total content of aldehydes in the U100 group (9570 μg/kg) was higher than that in the U0 group (7002 μg/kg), indicating that the accumulation of aldehydes was promoted by the addition of lipase. Benzaldehyde, pentanal, hexanal, heptanal, and octanal were significantly increased 1.1, 1.5, 1.4, 1.1 and 2.2 times, respectively, in the U100 group compared to the U0 group. These aldehydes were reported as the main compounds of the typical fresh fish flavor [21]. Benzaldehyde offers almond flavor and typical mushroom flavor [22]. Pentanal, hexanal and octanal have an odor similar to the aroma of fat and green [23,24]. Heptanal and octanal were considered to as important flavor compounds in fermented sausages [25]. Pentanal, hexanal, heptanal, and octanal were fatty aldehydes, which were oxidation products of unsaturated fatty acids. The addition of exogenous lipase could promote the release of free fatty acids (FFAs). The degradation and oxidation of lipids promotes the production of aldehydes, creating the characteristic fatty and grassy taste of *Suanzhayu*.

Esters have a low threshold and produce fruit and flower odors [26]. They were generally result from the esterification of short-chain acids and alcohols. Only two esters, ethyl ester hexanoic acid and ethyl ester octanoic acid, were detected in the samples. At the beginning of the fermentation (day 0), no esters were observed. As fermentation proceeded, esters started to appear at day 5 and gradually accumulated. At the end of fermentation, esters in the U100 group (276.1 μg/kg) were 11 times higher than those in the U0 group (24.7 μg/kg). Lipase catalyzes the hydrolysis of triglycerides et al. [27], producing large amounts of FFAs. With microbial esterase, FFA esterifies with alcohols to produce esters, such as ethyl ester hexanoic acid and ethyl ester octanoic acid in this study, providing a special fruity and ester flavors for the product [28]. These results suggested that the addition of lipase could promote the production of esters and promote the formation of fruity and ester odors in *Suanzhayu*.

The odor activity value was calculated to assess the contribution of volatile compounds to *Suanzhayu*. Thirteen volatiles with OAV > 1 were considered to contribute significantly to the characteristic flavor of the product (Table 1). Octanal had the largest OAV value. Octanal was reported to provide a grassy flavor to the product [24]. In addition, pentanal, hexanal, nonanal and 1-hexanol also have relatively large OAV values (OAV > 1500), and these substances may contribute to the formation of fat, grass, mushroom, and flower aromas in *Suanzhayu* [24,29]. The main odor active compounds were similar in both groups, indicating that the addition of lipase did not change the main odor profile of the product. In addition, the OAV values of most substances in the U100 group were higher than those in the U0 group, indicating that the addition of lipase elevated the concentration of volatile compounds (OAV > 1) that could be perceived, i.e., enhanced the flavor of *Suanzhayu*.

Hierarchical clustering analysis was performed with volatile compounds among different groups during the fermentation process and shown in Figure 3. The samples were divided into three categories: (1) 0d, U0–3d and U0–5d, the first stage of fermentation; (2) U0–7d and U100 group 3–5d, the second stage of fermentation; (3) U0–14d and U100 group 7–14d, the final stage of fermentation. The sample fermented for 7 days in the lipase addition group reached the flavor profile of the 14 d sample in the natural fermentation group, which could be considered as the maturation stage. Therefore, the addition of lipase might contribute to the formation of volatile substances. Similar reports had been reported in *Chouguiyu*, in which the volatile profile LW5 (inoculated with *Lactococcus lactis* M10 and *Weissella cibaria* M3 and fermented from 5 days 5) was the most similar to C7 (naturally fermented for 7 days) [10]. Therefore, the starter culture inoculation could promote the flavor formation in *Chouguiyu* products with time shortened. Ansorenaetal et al. also reported that the addition of lipase and protease reduced the fermentation time of dry sausages from 35 to 21 d [2]. Thus, lipase addition could be used as a method to promote the formation of volatile substances in *Suanzhayu*.

### 3.4. Correlation between Bacterial Community and Volatile Compounds, pH, and Lactic Acid Content

The correlations of the bacterial community with volatile compounds (OAV > 1), pH and lactic acid content were calculated and displayed in Figure 4. *Lactobacillus* showed a negative correlation with pH (R = −0.645) (*p* < 0.05) and a positive correlation with lactic acid content (R = 0.611) (*p* < 0.05), indicating that the accumulation of lactic acid and the decrease in pH were mainly caused by *Lactobacillus*. In contrast, *Escherichia-Shigella* had a positive correlation with pH (R = −0.849) (*p* < 0.01) and a negative correlation with lactic acid content (R = −0.653) (*p* < 0.05). *Escherichia-Shigella* was also found to be negatively correlated with *Enterococcus* (R = −0.619) (*p* < 0.05) and *Proteus* (R = −0.565) (*p* < 0.05), indicating that *Lactobacillus*, *Enterococcus* and *Proteus* had an inhibitory effect on the growth of *Escherichia-Shigella*, a typical spoilage bacterium [30]. *Lactobacillus*, *Enterococcus* and *Proteus* were important microorganisms for the safety of *Suanzhayu*, probably by producing acid and lowering the pH value, thus inhibiting of *Escherichia-Shigella*.

For flavor, *Lactobacillus*, *Enterococcus* and *Proteus* showed significant positive correlations with the 13 compounds (OAV > 1, Table 1), suggesting that these strains played an important role in the production of the characteristic compounds. These results were similar to the report in sourdough fermentation, that the group added with *Lactobacillus plantarum* had the highest content of heptanal [31]. In another study, 1-pentanol was produced during growth of all LAB species in sourdough [32]. In the fermented fish sausages, *Enterococcus* played a significant role in the formation of flavor characteristics [33], such as hexanal, heptanal, octanal, benzaldehyde, etc. *Proteus* species were reported to play important roles during cheese ripening [18], especially in the formation of flavor substances, such as 3-methyl-1-butanol, ethyl ester hexanoic acid, etc. Deetae et al. also found in the cheese model that *Proteus* was closely related to aldehyde content and played an important role in the formation of 3-methyl-1-butanol [18]. The above correlation analysis results showed that *Lactobacillus*, *Enterococcus* and *Proteus* played significant roles not only in product safety, but also in the formation of product flavor.

## 4. Conclusions

The addition of lipase had no significant effect on the pH of the *Suanzhayu*. The addition of lipase promoted the growth of *Proteus* and the formation of volatile substances (especially aldehydes and esters). The correlation analysis showed that *Lactobacillus*, *Enterococcus* and *Proteus* played an important role not only in the safety of the product but also in the formation of flavor. Addition of lipase could be used as a novel means to enhance the quality of *Suanzhayu*.

## Figures and Tables

**Figure 1 foods-10-02529-f001:**
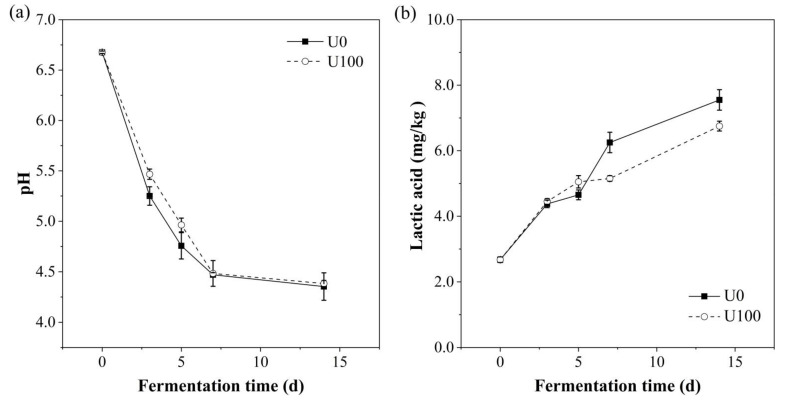
Changes of pH (**a**) and lactic acid content (**b**) in the samples during the fermentation of *Suanzhayu* without (U0) and with (U100) lipase. Samples were collected at 0, 3, 5, 7, and 14 d, respectively.

**Figure 2 foods-10-02529-f002:**
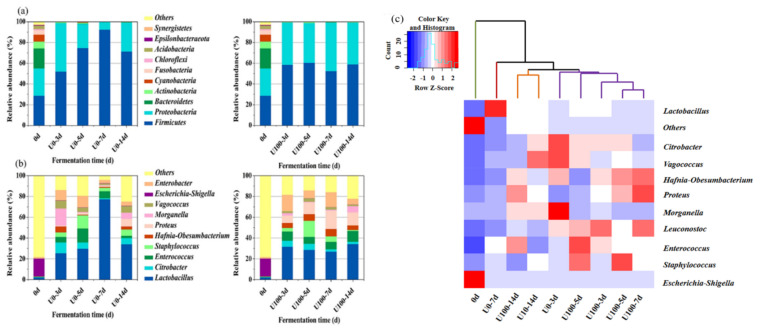
The relative abundances of bacteria at the phylum level (**a**) and the genus level (**b**) during the fermentation of *Suanzhayu* without (U0) and with (U100) lipase. (**c**) Heat map of the bacterial community compositions during the *Suanzhayu* fermentation without (U0) and with (U100) lipase. The colors indicate low (blue) to high (red).

**Figure 3 foods-10-02529-f003:**
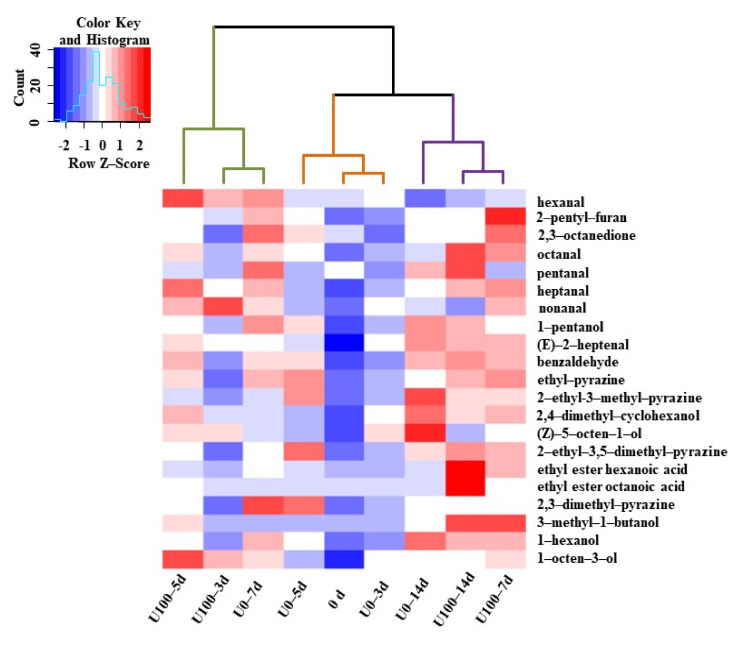
The heat map of volatile compositions during the fermentation *Suanzhayu* samples without (U0) and with (U100) lipase. The colors indicate low (**blue**) to high (**red**).

**Figure 4 foods-10-02529-f004:**
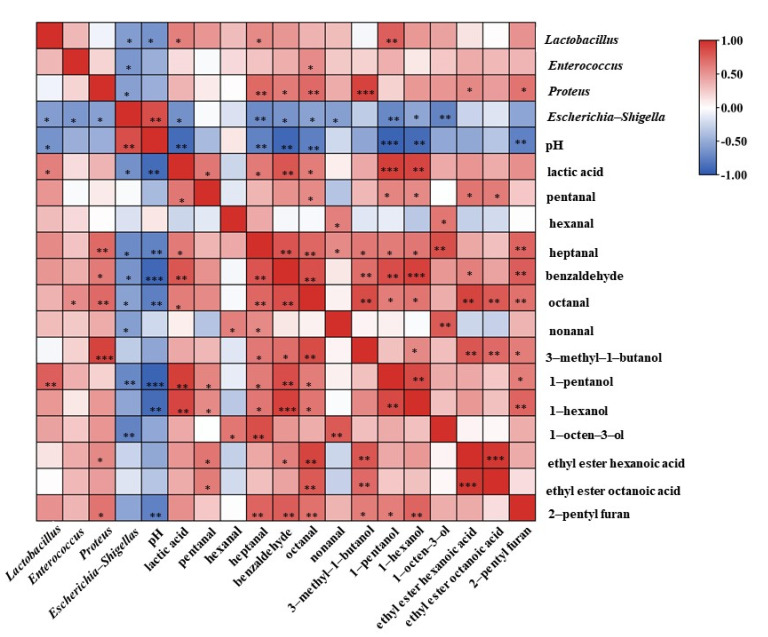
Correlations between the bacterial communities with volatile compounds (OAV > 1), pH and lactic acid in *Suanzhayu* samples. Red indicates positive correlations, while blue indicates negative correlations. Significant at: * (*p* < 0.05); ** (*p* < 0.01); *** (*p* < 0.001).

**Table 1 foods-10-02529-t001:** Concentration and OAV values of odor-active compounds in *Suanzhayu* (fermentation 14 day) with (U100) and without (U0) lipase.

Compound	Threshold Value	Content (μg/kg)		OAV*10		Significance
	(μg/Kg)	U0	U100	U0	U100	
pentanal	0.019	437.2	671.2	2300.8	3532.5	*
hexanal	3.3	4992.6	6931.7	151.3	210.1	*
heptanal	0.011	319.3	359.4	2902.6	3267.5	
benzaldehyde	0.85	230.8	253.5	27.2	29.8	
octanal	0.0005	387.0	839.3	77,397.7	167,855.4	**
nonanal	0.02	391.7	300.1	1958.5	1500.7	*
3-methyl-1-butanol	28	546.2	1903.9	2.0	6.8	
1-pentanol	3.6	303.6	259.2	8.4	7.2	
1-Hexanol	0.25	3938.3	2795.9	1575.3	1118.4	*
1-octene-3-ol	0.03	1829.7	1794.1	6776.7	6644.9	
ethyl ester hexanoic acid	1	24.7	199.9	2.5	20	**
ethyl ester octanoic acid	40	0.0	76.2	0.0	0.2	**
2-pentyl furan	6	447.9	438.3	7.5	7.3	

Note: * stands for significant difference (*p* < 0.05), and ** stands for extremely significant difference (*p* < 0.01).

## Data Availability

The data that support the findings of this study are available from the corresponding author upon reasonable request.

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
