# Peer review of "Lipase Addition Promoted the Growth of Proteus and the Formation of Volatile Compounds in Suanzhayu, a Traditional Fermented Fish Product"

_foods, 2021, doi:10.3390/foods10112529_

Round 1

Reviewer 1 Report

The manuscript entitled "Lipase addition promoted the growth of Proteus and the formation of volatile compounds in Suanzhayu, a traditional fermented fish product" describes a study in which the lipase is added into the original Chinese fermented product.

The manuscript is nice to read, introduction explains why the study is conducted, design and the description of the mentod are correct. The results and discussion section is easy to follow and explains signifficance of the study.

In general, very nice piece of work. Some minor comments, not affecting the overall quality are given below:

Line 47-48 - to big simplification. "... flavour such as butanoic acid..." is not correct phrase.

Line 114-116 - do you mean the oven temperature?

Author Response

Dear editor and reviewers: 
Thank you so much for your email of Sep 24, 2021, with the reviewers’ comments on our manuscript (ID: foods-1394574). We have studied comments seriously and have made corrections accordingly which we hope to meet with approval. Revised portions are marked in red in the paper, and some are further explained below as “Response to editor and reviewers’ comments”. Please see the attachment.

Reviewer 2 Report

General comments:

The manuscript deals with the study on the effects of lipase addition on pH, lactic acid content, and volatile components during the fermentation of Suanzhayu, a Chinese traditional fermented fish product.

The paper is well-written and could be interesting for a wide group of readers. Although the methodology used is correct, some important results should be appropriately discussed and clarified. In the Results and discussion section, the discussion of some results is speculative and confuse, being very difficult to understand.

Other considerations are as follows:

2. Materials and methods section:

2.1. Subsection: Preparation of Suanzhayu samples. Lines 67-69: Why was a ratio of 100 U of lipase/g fish meat used in the Suanzhayu samples group U100?

2.2. Line 73: Why were the samples stored at -20 ºC until subsequent analysis within two months?

3.   Results and Discussion section.

3.1. Subsection: Changes of pH and lactic acid contents during fermentation. Lines 135-140: The culture pH in both the fermented U0 and U100 samples evolved in a similar way and reached the same final pH value (4.4). However, lactic acid production exhibited a similar profile in the fermented U0 and U100 samples, but the final lactic acid concentrations were significantly different: 7.6 mg/kg and 6.8 mg/kg in U0 and U100 groups, respectively. This result is rare since the authors did not detect, in both fermented samples, the production of other organic acids in addition to lactic acid (right part of Figure 1). So that, if both fermented samples reached the same final pH (left part of Figure 1), the final concentrations of lactic acid should also be very similar. However, this was not observed. The authors should explain clearly this contradictory result.

3.2. Subsection: Changes of pH and lactic acid contents during fermentation. Lines 140-141: This affirmation is incorrect since the content of lactic acid does not increase with the decrease of pH. Actually, the culture pH decreases with the increase in the concentration of lactic acid.

3.3. Lines 141-143: This affirmation is repetitive. This sentence should be rewritten as follows: This accumulation of lactic acid during the fermentation process can inhibit the growth of spoilage and pathogenic bacteria, thereby extending the shelf life of the product.

3.4. Lines 145-146: This affirmation is repetitive and should be deleted.

3.5. Lines 147-148: I do not understand this affirmation. Which expected trend do the authors refer to?

As observed in Figure 1, the pH time-course in the samples during the fermentation of Suanzhayu without (U0) and with (U100) lipase exhibited the commonly observed decrease during fermentation processes.

3.6. Lines 148-151: This sentence is so repetitive and confusing. Was the production of other organic acids (acetic and propionic acids) acid not detected in the fermented samples? Why?

It is not clear how the production of lactic acid and other organic acids (acetic and propionic acids) can change the dynamics of the culture pH. So that, these affirmations (lines 147-152) should be rewritten because to avoid confusion to the readers.   

  1. Section: Conclusion. Lines 323-324: The authors stated: “The addition of lipase had no significant effect on the pH and lactic acid content of the Suanzhayu samples.”

However, from detailed observation of Figure 1, it can be noted that the addition of lipase did not significantly affect the pH evolution during fermentation, but the production of lactic acid depended on the enzyme addition. This fact should be appropriately discussed and clarified in the Results and Discussion section, and after this, the Conclusion section should be rewritten according to the results obtained.  

Author Response

(The authors gave the same response as above.)

Round 2

Reviewer 2 Report

Dear editor and reviewers:

Thank you so much for your email of Sep 24, 2021, with the reviewers’ comments on our manuscript (ID: foods-1394574). We have studied comments seriously and have made corrections accordingly which we hope to meet with approval. Revised portions are marked in red in the paper, and some are further explained below as “Response to editor and reviewers’ comments”.

Reviewers Comments:

To Reviewer 2:

Comments to Author

The paper is well-written and could be interesting for a wide group of readers. Although the methodology used is correct, some important results should be appropriately discussed and clarified. In the Results and discussion section, the discussion of some results is speculative and confuse, being very difficult to understand.

Other considerations are as follows:

Materials and methods section:

Question 1: Subsection: Preparation of Suanzhayu samples. Lines 67-69: Why was a ratio of 100 U of lipase/g fish meat used in the Suanzhayu samples group U100?

Reply 1: Thank you for your comment.

The lipase addition was based on the following reasons. First, the lipase activity (PANGBO ENZYME, Nanning, China) is 100,000 U/g, and the recommended dose is 0.1 to 0.3%, i.e., 100 U/g to 300 U/g. Second, by reviewing the literatures, we found that 100 U/g to 300 U/g is a common addition in fermented meat products. For example, Rani and Jagtap reported that 200 U/g of lipase was added to cheese, and the cheese maturation time was shortened from 90 days to 60 days while maintaining its quality (Journal of Food Science and Technology, 2019, 56(1): 497-506.). Finally, different lipase addition at 100 U/g, 200 U/g and 300 U/g were prepared, and the pre-experiment showed that the effect of adding 100 U/g fish meat of lipase was of good acceptance. Therefore, we decided to add 100 U/g lipase in Suanzhayu.

Reviewer response: Please include the latter paragraph in the text (Subsection: Preparation of Suanzhayu samples) to explain why was a ratio of 100 U of lipase/g fish meat used in the Suanzhayu samples group U100.

 Question 2: Line 73: Why were the samples stored at -20 ºC until subsequent analysis within two months?

Reply 2: To be precise, samples for volatile compounds and microbiological analysis were kept at -80 ºC and samples for other analysis were stored at -20 ºC. We have added these details on Lines 90-92.

The purpose is that, first of all, it was not possible to perform the analysis immediately after sampling on the same day, so we stored the samples in the refrigerator for the subsequent analysis. In addition, to prevent quality changes during the freezing process, such as fat oxidation, we complete the analysis of all samples within 2 months to ensure the accuracy of the results.

Reviewer response: Ok.

Results and Discussion section:

Question 3: Subsection: Changes of pH and lactic acid contents during fermentation. Lines 135-140: The culture pH in both the fermented U0 and U100 samples evolved in a similar way and reached the same final pH value (4.4). However, lactic acid production exhibited a similar profile in the fermented U0 and U100 samples, but the final lactic acid concentrations were significantly different: 7.6 mg/kg and 6.8 mg/kg in U0 and U100 groups, respectively. This result is rare since the authors did not detect, in both fermented samples, the production of other organic acids in addition to lactic acid (right part of Figure 1). So that, if both fermented samples reached the same final pH (left part of Figure 1), the final concentrations of lactic acid should also be very similar. However, this was not observed. The authors should explain clearly this contradictory result.

Reply 3: Indeed, both the fermented U0 and U100 samples reached the same final pH (4.4), however, their final lactic acid concentrations were different. “ The reason for this may be due to the fact that pH value is not only influenced by lactic acid but also other organic acids, such as acetic acid and propionic acid from some acidifying bacteria (Juárez-Castelán et al., 2019).”  We supplemented the information on Line 166-168.

The phenomenon is common in other studies. For example, Liu et al. (Food Chemistry, 2018, 266: 262-274.) reported that the pH of the TD705261 group (bilberry wine inoculated with Torulaspora delbrueckii 70526) and the group TD291 (bilberry wine inoculated with Torulaspora delbrueckii 291) were both 3.58, with no significant difference. However, there was a significant difference in lactic acid content between the two groups, 118.69 mg/L and 100.09 mg/L. The reason may be the difference in pyruvic acid content. Pyruvic acid was 42.9 mg/L in the TD705261 group, which was lower than in the TD291 group (69.62 mg/L). Another research on fermented sausage supplemented with pineapple (Food science and biotechnology, 2016, 25(6): 1657-1664.) showed that the pH of both the control and 2% pineapple sausage groups was at 4.5, with no significant difference. However, their lactic acid content had a significant different (P<0.05) at 100mg/100g and 150mg/100g, respectively. This was due to the increase of other organic acids (phosphoric acid, succinic acid and glutamic acid) in the control group during the fermentation process, which eventually led to the same pH value in both groups. In a study on fermented fish, Zeng et al. (Food Control, 2013, 30(2): 590-595) reported that, in addition to lactic acid, higher levels of acetic acid in fermented fish samples contributed to improved organoleptic acceptability. Therefore, it is not difficult to understand that in our study there was a difference in the final lactic acid between the two groups at similar pH values. The reason may be that, in addition to lactic acid, organic acids in fermented products include other organic acids such as pyruvic acid, acetic acid, succinic acid and so on, which may also have an effect on the pH value.

Reviewer response: I understand the response given by the authors based on the different references cited by them in this response. However, how was possible that other organic acids different from lactic acid were not detected by HPLC. It is a rare case, since this technique allows the determination of different organic acids and sugars. Therefore, the authors should have analyzed their samples again to detect the other organic acids affecting the final pH value of the samples to clearly explain the results obtained. So that, it is difficult to know which organic acid is responsible for the same pH evolution and final pH value and the different lactic acid production in the fermented U0 and U100 samples. In summary, the explanation given by the authors to explain this fact is speculative. So that, this question remains unclear.

Question 4: Subsection: Changes of pH and lactic acid contents during fermentation. Lines 140-141: This affirmation is incorrect since the content of lactic acid does not increase with the decrease of pH. Actually, the culture pH decreases with the increase in the concentration of lactic acid.

Reply 4: Thank you for your correction. The previous words “The content of lactic acid increases with the decrease of pH” have been revised to “The culture pH decreases with the increase in the concentration of lactic acid” on Line 155-156.

Reviewer response: Ok.

Question 5: Lines 141-143: This affirmation is repetitive. This sentence should be rewritten as follows: This accumulation of lactic acid during the fermentation process can inhibit the growth of spoilage and pathogenic bacteria, thereby extending the shelf life of the product.

Reply 5: Thank you for your correction. The previous words “This is due to the accumulation of organic acids during the fermentation process, which can inhibit the growth of spoilage bacteria and pathogenic bacteria, thereby extending the shelf life of the product” have been revised to “This accumulation of lactic acid during the fermentation process can inhibit the growth of spoilage and pathogenic bacteria, thereby extending the shelf life of the product.” on Line 156-157.

Reviewer response: Ok.

Question 6: Lines 145-146: This affirmation is repetitive and should be deleted.

Reply 6: Thank you for your comment and we have removed the previous sentence: “It was reported that lactic acid bacteria could ferment carbohydrates into energy and lactic acid, resulting in a pH decrease (Swetwiwathana & Visessanguan, 2015).”

Reviewer response: Ok.

Question 7: Lines 147-148: I do not understand this affirmation. Which expected trend do the authors refer to?

As observed in Figure 1, the pH time-course in the samples during the fermentation of Suanzhayu without (U0) and with (U100) lipase exhibited the commonly observed decrease during fermentation processes.

Reply 7: Thank you for your comment. We have removed misleading statements.

Reviewer response: Ok.

Question 8: Lines 148-151: This sentence is so repetitive and confusing. Was the production of other organic acids (acetic and propionic acids) acid not detected in the fermented samples? Why?

It is not clear how the production of lactic acid and other organic acids (acetic and propionic acids) can change the dynamics of the culture pH. So that, these affirmations (lines 147-152) should be rewritten to avoid confusion to the readers.

Reply 8: Thank you for your comment. We did not test for organic acids other than lactic acid because when we used high performance ion chromatography (Dionex ICS-5000 + DC, Thermo Scientific, USA) to establish the method, we only have standards for lactic acid. Indeed, as you said, other organic acids, such as acetic acid and propionic acid, are also very important, which have an important impact on fermented foods. We will consider establishing methods for the detection of other organic acids. Thank you for your suggestion.

We have revised the confusing statement, as following: “However, at the end of fermentation, there was a significant difference in lactic acid between the two groups, while the pH values were the same. The reason for this may be due to the fact that pH value is not only influenced by lactic acid but also other organic acids, such as acetic acid and propionic acid from some acidifying bacteria (Juárez-Castelán et al., 2019). In all, results indicate that the addition of lipase had no effect on the pH of Suanzhayu, but caused a decrease in the lactic acid content.” on line 164-169.

Reviewer response: see reviewer comment to Question 3.

Conclusion section:

Question 9: Lines 323-324: The authors stated: “The addition of lipase had no significant effect on the pH and lactic acid content of the Suanzhayu samples.” However, from detailed observation of Figure 1, it can be noted that the addition of lipase did not significantly affect the pH evolution during fermentation, but the production of lactic acid depended on the enzyme addition. This fact should be appropriately discussed and clarified in the Results and Discussion section, and after this, the Conclusion section should be rewritten according to the results obtained. 

Reply 9: Thank you for your comment. In the previous conclusion section, the addition of lipase had no significant effect on the pH and lactic acid content of the Suanzhayu samples. These speculations are deleted.

The related conclusion part is rewritten into: “The addition of lipase had no significant effect on the pH of the Suanzhayu, but caused a decrease in the lactic acid content.” on Line 312-313.

We have added explanations in the part of results and discussion.“However, the lactic acid content of the U100 group at the end of fermentation was lower than that of the U0 group. Sun (Sun et al., 2010) and Knez (Knez et al., 2012) reported that lipase can catalyze the esterification of lactic acid and other fatty acids with alcohol. Therefore, the lower lactic acid content in U100 group might be due to the consumption of lactic acid by lipase-catalyzed esterification. This may also provide the possibility of flavor enhancement in the enzyme addition group. In most studies, pH was negatively correlated with lactic acid, and this phenomenon was also observed in our study. However, at the end of fermentation, there was a significant difference in lactic acid between the two groups, while the pH values were the same. The reason for this may be due to the fact that pH value is not only influenced by lactic acid but also other organic acids, such as acetic acid and propionic acid from some acidifying bacteria (Juárez-Castelán et al., 2019). In all, results indicate that the addition of lipase had no effect on the pH of Suanzhayu, but caused a decrease in the lactic acid content.”These are supplemented on line 158-169.

Reviewer response: see reviewer comment to Question 3.

Author Response

Dear editor and reviewers:

Thank you so much for your email of Oct 07, 2021, with the reviewers’ comments on our manuscript (ID: foods-1394574). We have studied comments seriously and have made corrections accordingly which we hope to meet with approval. Revised portions are marked in red in the paper, and some are further explained below as “Response to editor and reviewers’ comments”.

The details are shown in the attechment.
